# Generating Multi-turn Clarification for Web Information Seeking

## ABSTRACT

Asking *multi-turn* clarifying questions has been applied in various conversational search systems to help recommend people, commodities, and images to users. However, its importance is still not emphasized in Web search. In this paper, we make the first attempt to extend the multi-turn clarification *generation* to *Web search* for clarifying users' ambiguous or faceted intents. Compared with other conversational search scenarios, Web search queries are more complicated, so the clarification should be *generated* instead of *selected* that is commonly applied in existing studies. To this end, we first define the whole process of multi-turn Web search clarification composed of clarification candidate generation, optimal clarification selection, and document retrieval. Due to the lack of multi-turn open-domain clarification data, we first design a simple yet effective rule-based method to fit the above three components. After that, by utilizing the in-context learning and zero-shot instruction ability of large language models (LLMs), we implement clarification generation and selection by prompting LLMs with a few demonstrations and declarations, further improving the clarification effectiveness. To evaluate our proposed methods, we first apply the Qulac dataset to measure whether our methods can improve the ability to retrieve documents. We further evaluate the quality of generated aspect items with MIMICS dataset. Experimental results show that, compared with existing single-turn methods for Web search clarification, our proposed framework is more suitable for open-domain Web search systems in asking multi-turn clarification questions to clarify users' ambiguous or faceted intents.

## 1 INTRODUCTION

Search clarification has become an important part of open-domain conversational Web search [4, 43, 44]. According to the existing definition, when a user issues an ambiguous or faceted Web query, the system asks the user a *clarifying question* and provides several *candidate items* (attributes) representing potential intents for the user to select [42]. The query can be refined according to the user selection to retrieve a new list of documents (passages), and the system can continue to clarify until the user intent is deemed specific enough. A typical clarification process is shown in Figure 1. It can be seen that the Web search clarification process is essentially a *multi-turn* interaction or conversation between the user and the system. The multi-turn mechanism is especially emphasized when the user's search intent is complicated or less specific, while single-turn clarification cannot satisfy the user's need [5, 22, 35].

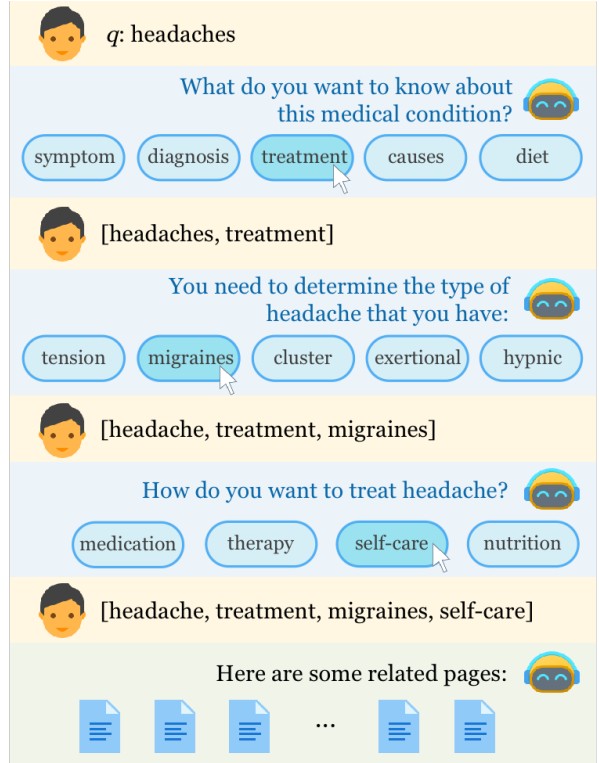

**Figure 1: A process of asking multi-turn clarifying questions. When the system deems that the user's intent is still ambiguous or faceted, it generates a clarification pane. The process can repeat for several turns until the query intent is clear.**

Nowadays, multi-turn clarification has been applied in many Information Retrieval (IR) scenarios [25]. For example, in Conversational Recommender Systems (CRS), the system asks the user about the attributes of commodities turn-by-turn for recommendation [6, 18, 19, 49]. In conversational search systems, the system asks the user to deliver more information about her needs [1, 10, 15]. Besides, multi-turn clarification has also been applied in other close-domain applications like interactive classification [41] and twenty-questions task for picture guessing [39]. These studies discuss the importance of multi-turn clarification in IR systems, and inspire us that open-domain Web search should also include such a *multi-turn* process to help users better find the information they want.

However, existing Web search clarification studies [30, 36, 42–44, 47] focus on generating a single pane [11, 12, 26], and continuing the process by restarting a clarification based on the updated query to achieve a *pseudo* multi-turn process. Although this approach can generate reasonable clarification panes, it is not the optimal choice in a multi-turn scenario, which makes it **deviate from the user's intent**, or requires **lots of turns** to find the user's intent. This is because existing methods only adopt greedy strategies and cannot consider the global relation of multi-dimensional potential

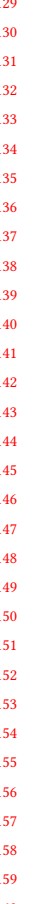
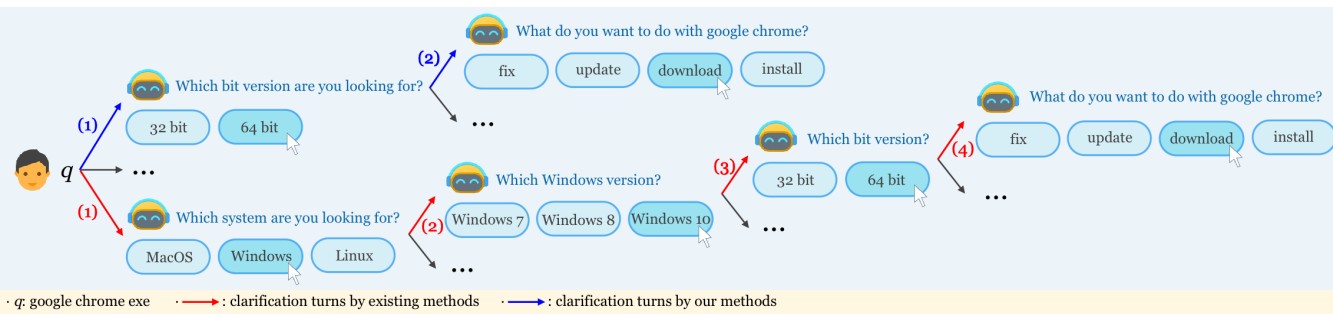

**Figure 2: Comparison of different clarification paths. In this paper, we emphasize the importance of satisfying the users' information need with as-short-as-possible path (clarification turns).**

panes in each turn. For example in Figure 2, we assume that the user intent "download google chrome exe 64 bit" can be achieved by different clarification paths. Since existing methods often ignore the potentially multi-dimensional essence at each turn, they can easily go through some wrong or long paths (the red one). **In this paper, we make the first step to try to extend existing multi-turn clarification scenario into the Web search** by *generating* several potential clarification panes at each turn and select the optimal path (the blue one in Figure 2) to achieve multi-turn Web clarification.

To formally describe the problem we are studying, we refer to existing close-domain clarification scenarios, especially CRS, and identify a main goal and a framework composed of three main components. Our main goal is to **help users search for satisfying documents with as few clarification turns as possible**. To achieve this, we formalize the multi-turn process, and then define a framework including three components: (1) *Clarification candidate generation*: Since the target (documents) to be recommended in Web search is dynamically updated with each turn of user selection, it is necessary to dynamically generate potential clarification panes in each turn. (2) *Optimal clarification selection*: Similar to CRS, each query may have multiple potential clarification panes. After the clarification candidate generation, a certain strategy needs to be applied to select the optimal clarification pane composed of a question and several clickable items. (3) *Document retrieval*: After the user submits the query or clicks one item for clarification at each turn, a new document list should be retrieved.

Due to the lack of corresponding studies and datasets, we first design a rule-based method **MulClari-Rule**, to fit our proposed framework. The method (1) first finds candidate items from search result pages and clusters them to construct the candidate items set, then generates a question based on existing question generation algorithms [42, 47], and (2) selects the optimal clarification at each turn by applying *max-entropy strategy*. We retrieve documents using BM25. However, the rule-based method is weak at obtaining abundant contextual information between different turns. Nowadays, large language models (LLMs) have performed well in many Natural Language Processing (NLP) tasks. Their zero-shot instruction and in-context learning ability lead to strong ability in multi-turn conversation modeling. With their abilities, in this paper, we further propose an LLM-based method **MulClari-LLM** prompted by human-designed demonstrations, to implement the components above and achieve multi-turn Web search clarification.

It is challenging to evaluate multi-turn Web search clarification due to the diversity of user queries. In this paper, we propose evaluating the multi-turn clarification from two perspectives: (1) Since our goal is to provide users with documents in as-few-as-possible clarification turns, we first evaluate the ability to retrieve satisfying documents for ambiguous or faceted queries. To achieve this, we rely on the Qulac dataset [1] together with its relevance judgements and evaluate the document ranking results after clarification by MRR, NDCG, MAP, and P@1. (2) We also evaluate the quality of the first-turn generated clarification panes using single-turn evaluation metrics based on the MIMICS dataset [43]. The experimental results demonstrate that, first, compared with single-turn baseline models, our proposed multi-turn strategy can find the information the user wants in as few turns as possible. Second, the goals of multi-turn scenarios are significantly different from those of single-turn scenarios, and the definition of high quality in single-turn scenarios may not necessarily meet the goals of multi-turn scenarios.

The main contributions of this paper include:

- To our best knowledge, we are the first to extend the single-turn Web search clarification to multi-turn, enriching the existing conversational search scenarios.
- We define the goal of Web search clarification and the generation process including clarification candidate generation, optimal clarification selection, and document retrieval. We further design a rule-based and an LLM-based method to implement our proposed framework.
- We design two approaches to evaluate our proposed methods. The experimental results show that our methods can retrieve satisfying documents in as few turns as possible.

## 2 RELATED WORK

***Open-domain Search Clarification.*** Aliannejadi et al. [1] first proposed asking clarifying questions in conversational search systems. However, it can only retrieve and select questions and let the user respond by natural language, which is not suitable for Web search. In Web search systems, the user's query is very complex, so the question selection is extremely difficult, because the question in the limited dataset can not satisfy the large-scale Web query. More recently, Zamani et al [42, 44] first emphasize the importance of clarification in the field of Web information retrieval. They propose the MIMICS [30, 43] for the Web search clarification. In Web search

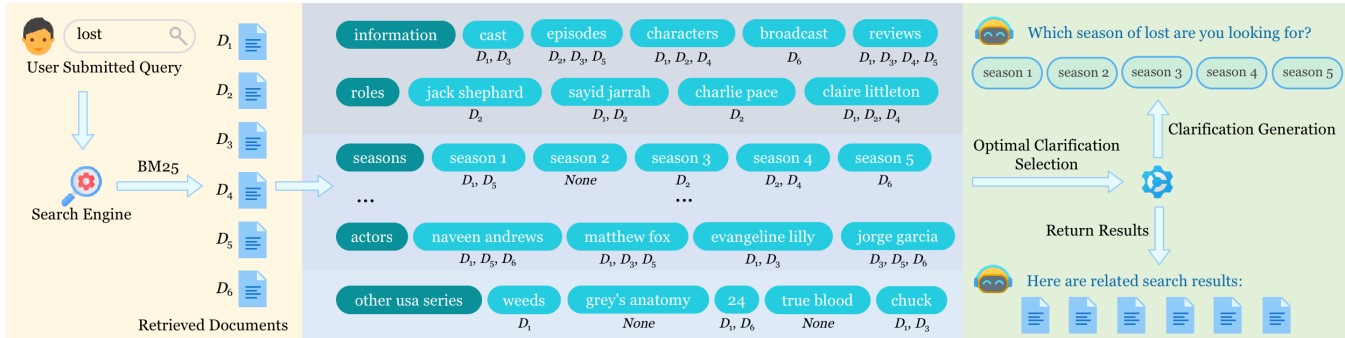

**Figure 3: Our proposed framework taking the query "lost" as an example. It is composed of three main components: (1) Clarification candidate generation, (2) Optimal clarification selection, and (3) Document Retrieval.**

clarification, the clarifying question [37] together with the aspect items [8, 13, 14] are generated instead of selected in conversational search systems [1] or constructed by some rules in CRS [18], ensuring its essence of open-domain. Besides, some close-domain clarification generation methods [23, 24, 33, 40] also show their strength in some question answering communities, yet they also cannot cover a wide range or Web search queries.

***Conversational Recommender System (CRS).*** CRS aims at mining users' preferences through multiple turns of natural language conversation, so as to recommend to users the items they may be interested in. Sun and Zhang [29] first proposed the concept of CRS, and considered its several important issues. These problems show various similarities with search clarification systems [44]. Later, researchers have tried various algorithms to perform conversational recommendation [3, 6, 16, 18, 19, 41, 45, 48, 49]. It is worth noticing that, in CRS, each commodity to be recommended has its own attribute set, such as the brand and CPU model of the computer. Therefore, it is an important step to select an attribute from them to ask the user. In this process, some strategies like max-entropy have been widely applied, or they can be achieved by applying more deep-inside natural language understanding models. However, different from CRS, in Web search, unlike static commodities, retrieved documents are real-time-updated with a huge quantity, and do not have fixed attribute sets, so it is necessary to dynamically *generate* the attribute set for each document.

***Other Clarification Scenarios.*** In addition to the main-stream open-domain and close-domain search clarification scenarios mentioned above, search clarification can also be applied in some other novel scenarios. For example, Yu et al. [41] studied how to classify objects in an interactive way. They gradually determine what users are thinking by asking them multiple turns of questions to clarify their intents in a multiple-choice manner. White et al. [39] proposed a novel scenario: guessing which image the user has in mind within 20 Yes/No questions. It also borrows some ideas from CRS. Zhang and Zhu [46] studied about what information was omitted when publishing products on e-commerce websites, and informed publishers in the form of questions. Recently, Shi et al. [28] studied whether a certain step in game intelligence should be taken or a question asked of the user to resolve ambiguity. These studies provide potential application scenarios for search clarification.

## 3 MULTI-TURN WEB CLARIFICATION

### 3.1 The Necessity of Multi-turn Clarification

Existing studies of Web search clarification focus on generating high-quality clarifying questions and aspect items (or attributes) given a user query in a single-turn setting. However, when the user intent is complex, single-turn clarification is not the best choice to satisfy the complicated user intent because single-turn methods are not aware of global potential clarification candidates, thereby making it easy to go through wrong paths in potential clarification distributions as shown in Figure 2. By applying multi-turn clarification, the system can gradually clarify the user's complicated search intent turn-by-turn to improve their search efficiency and experience. For example, in CRS, the system asks the user about the attributes of commodities turn-by-turn for recommendation [6, 18, 19, 49]. In conversational search systems, the system asks the user to let the user deliver more information about her needs [1]. Besides, multi-turn clarification has also been applied in other close-domain applications like interactive classification [41] and twenty-questions task for image guessing [39]. These application scenarios provide inspiration for us to expand multi-turn search clarification to the Web search.

In Web search, the situation is more complex. First, Web search queries are *open-domain*, covering all kinds of real-world intents. This makes it sometimes difficult to understand the user intent and emphasizes the very importance of multi-turn clarification compared with other close-domain scenarios such as CRS. Therefore, the clarification pane should be *generated* instead of *selected or constructed by some rules*. Second, the item to be recommended in Web search is *large-scale documents or natural language passages* instead of a set of people, images, or commodities with clear attributes. It is difficult to represent a document or passage using existing attribute-based approaches. Therefore, it also emphasizes the necessity for mining attribute sets for a specific document.

### 3.2 Problem Reformulation

To solve the problem of multi-turn Web search clarification, we need to first define and formulate this task. (1) **First**, the user submits a query $q$, her target is relevant document set $D^q$. (2) **Next**, the system interacts with the user with multi-turn clarification: The system provides a clarification pane $C_i = (Q_i, S_i)$, and then the user

selects a candidate aspect item $A_i = (q_i, D_i)$. After that, the system updates documents based on $A_i$ and generates a new clarification pane $C_{i+1}$. (3) **Finally**, after the above $k$ turns, we calculate the performance of retrieving documents from a large document set. The whole process can be represented formally:

$$
\begin{aligned}
& q, C_1, A_1, C_2, A_2, \cdots, C_k, A_k, \Phi \\
=& q, (Q_1, S_1), A_1, (Q_2, S_2), A_2, \cdots, (Q_k, S_k), A_k, \Phi \\
=& q, (Q_1, S_1), (q_1, D_1), \cdots, (Q_k, S_k), (q_k, D_k), \Phi
\end{aligned}
\tag{1}
$$

Our task is to retrieve the documents (passages) satisfying the user's information need within a few clarification turns. It is worth noticing that the formulation is similar to the multi-turn clarification in conversational search systems [1, 10]. However, in conversational search systems, first, the user can only respond to the system by inputting a new sentence of natural language, which is time-wasting and experience-effecting. In our scenario, the user can respond just by clicking a candidate attribute staying consistent with existing single-turn Web search clarification [42–44], which is convenient for the user. Second, since Web search is complicated, we should **generate** instead of **selecting** clarification panes. In fact, existing studies in conversational search systems focus on selecting clarification panes from a question bank [1, 10], lacking universality for different queries. In contrast, for each specific query, we borrow the idea from CRS by generating various clarification panes according to the query and selecting the optimal one.

### 3.3 Framework Overview

To achieve our main goal, we design a framework containing three components, including: (1) *Clarification candidate generation*: Unlike static attribute sets in existing studies (such as CRS), the target to be recommended in Web search is dynamically updated with each turn of the item selected by the user. Therefore, it is necessary to generate multi-dimensional clarification pane candidates to have a global perspective of all potential panes. (2) *Optimal clarification selection*: Similar to CRS, we need to select the optimal clarification pane to deliver to the user based on some strategies. (3) *Document retrieval*: Retrieving relevant documents based on the user query and user-selected items. To implement the above three components, we first design a rule-based method MulClari-Rule. This method uses matching and human-designed features to extract candidate clarification panes from retrieved documents, and then select the optimal pane using the maximum information gain (or max-entropy, the same as below) strategy. Since the rule-based method makes it difficult to capture multi-turn semantic information, we further design another method MulClari-LLM leveraging the strong natural language understanding and generation ability of LLMs [2, 9, 21, 31, 32] to implement the clarification generation and selection process.

### 3.4 MulClari-Rule

*3.4.1 Rule-based Clarification Candidate Generation.* For a query $q$, we first obtain its corresponding potential multiple sets of candidate items as shown in the middle part of Figure 3. This is done to allow the system to be aware of all potential attributes of the query, making it convenient for our method to select the best item dimension from them, which is also consistent with existing systems such as CRS [18]. To obtain the multi-dimensional clarification candidates,

we designed a method MulClari-Rule that combines a generative model and well-designed manual rules. This method consists of three steps: (1) First, we need to generate an **independent item candidate set** $I^c$ containing many individual items that do not have group relations. (2) Then, since we deem that high-quality aspect items can be found in search result documents, which have been approved by some previous studies [42, 47], we only select the items that have appeared in the corresponding documents of the query as $I^s$, thereby **filtering out** some low-quality or wrongly-generated items. (3) Finally, since we need item dimensions divided by groups as shown in Figure 3, we **cluster the items** using co-occurrence information from MIMICS and select high-quality item dimensions and high-quality items in each dimension as the final result.

For the first step, we use BART [17] to generate independent candidate aspect items relying on its strong natural language generation ability. We first collect data pairs from MIMICS dataset [43] denoted as $(q, D) \rightarrow S_i$, where $q$ is the user query, $D$ is the top-10 search snippets, and $S_i$ is one item for a query. A query in MIMICS corresponds to up to five items. In order to provide sufficient candidates, we use beam search to take the first 100 beams of items generated by BART as the preliminary item candidate set $I^c$:

$$
I^c = \text{beam\_search}_{100}(\text{BART}(q, D))
\tag{2}
$$

where $q$ and $D$ are the same as above. These two are usually concatenated for aspect item generation [26]. After that, we delete items in $I^c$ that do not appear in $D$ to obtain a selected items candidate set $I^s$, to ensure the quality of the items.

Since each item in $I^s$ is independent without grouping, we need to cluster related items together to construct multi-dimensional items. For example in Figure 3, for the query "lost", it can generate five item dimensions, including (1) the information of this series, (2) roles, (3) seasons, (4) actors, and (5) other USA series. **The items in one dimension show high correlation**. To achieve this, we can rely on the co-occurrence information in the MIMICS dataset [43] and build a graph $G^M = <V^M, E^M>$ containing the co-occurrence frequency. In the graph, one node $V_i^M$ means one item in the MIMICS dataset, and one edge $E_{ij}^M$ represents the co-occurrence frequency between the item $V_i^M$ and $V_j^M$. Then, we cluster the generated items candidate set $I^s$ based on $G^M$. Specifically, we initialize a new graph $G^I$. For two items $I_i^s$ and $I_j^s$, if they are in $V^M$ and $E_{ij}^M$ exists, then we add the two nodes into $G^I$ as $V_i^I$ and $V_j^I$, and then build an edge $E_{ij}^I$ between these two nodes. Finally, we take out all $k$ fully connected components in $G^I$ as $k$ dimensions of generated items, sort the items in each dimension in descending order of their frequency in $G^I$, and select the top-5 items with the highest frequency in each dimension as the results, staying consistent with that in MIMICS dataset.

We conducted additional processes to ensure the quality of the generated items. First, in order to avoid repeating clarifications in multiple turns, we record the items presented to users in history and delete the clarification candidates containing these items in subsequent generations. In addition, after using BART for single-item generation, we use the part-of-speech analysis tool Stanza [20] to convert all plural items into singular and perform deduplication.

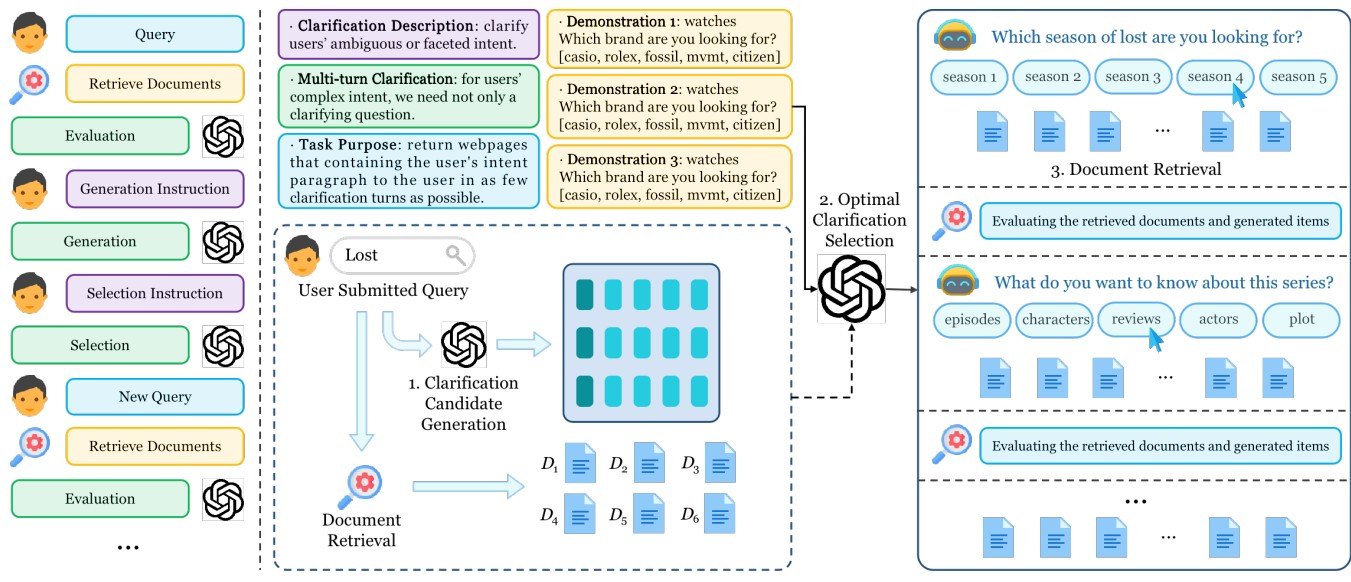

**Figure 4: Our proposed LLM-based method MulClari-LLM.**

Besides, given the user query $q$ and corresponding items $S$, we can apply some robust clarifying question generation algorithms [27, 38, 42, 47] to generate a question $Q$: $(q, S) \rightarrow Q$, to form a whole clarification pane as shown in Figure 1. For a specific query, several panes could be generated as clarification candidates.

*3.4.2 Rule-based Optimal Clarification Selection.* After generating multi-dimensional clarification candidates, we need to select the optimal pane to deliver to the user. To our best knowledge, in some systems like CRS [6, 19, 39, 45], max-entropy (ME, or max information gain) strategy has been widely applied for selecting the optimal attribute [39]. On the base of ME strategy, we also design an optimal clarification selection strategy. We aim to select the clarification pane that can mostly partition the retrieved documents. Specifically, for one dimension of items, we first list which documents each item appears in as shown in the middle part of Figure 3. We then calculate the information gain $Gain(\cdot)$ for each item $s$ as:

$$Gain(s) = H(D|q) - H(D|q, s) \quad (3)$$

where $q$ is the user query, $s$ is one item, and $D$ is the document set. Due to the large number of candidate documents, we only selected the top 50 documents retrieved by BM25 as the candidates. $H(D|q)$ is set to be 1, and $H(D|q, s)$ is the rate of the documents that containing $s$ in top-50 documents of $q$.

We further define the information gain of a dimension of items $S_i$ as the average information gain for each item in this dimension:

$$Gain(S_i) = \frac{1}{|S_i|} \sum_{s \in S_i} Gain(s) \quad (4)$$

Finally, we select the dimension with the highest information gain to obtain the optimal clarification.

$$S^o = \arg\min_{S_i \in S} Gain(S_i) \quad (5)$$

The selected dimension of items together with the generated corresponding clarifying question are then delivered to the user.

*3.4.3 Document Retrieval.* We implement and apply the BM25 algorithm to retrieve relevant documents of the query ("lost" for example) and return the newly generated clarification pane and the retrieved documents to the user. When the user clicks one of the provided items ("episodes for example in Figure 3), the query will be updated by concatenating the original query and the clicked item ("lost episodes" for example) to retrieve a new document list.

### 3.5 MulClari-LLMs

In MulClari-Rule, we basically rely on the frequency information of the items to generate multi-dimension clarifications, and then select the optimal clarification with max-entropy strategy. However, it has two limitations: **First**, it is still not good at **modeling multi-turn context**. When it selects the optimal clarification at each turn, it just focuses on maximizing the information gain, yet would not consider the context before due to the essence of the max-entropy strategy. **Second**, we assume that high-quality items should occur in top-retrieved document, but it cannot cover all potential high-quality items [26]. Recently, LLMs have performed well in various NLP tasks due to their strong in-context learning and zero-shot instruction ability. The natural language modeling ability of LLM can essentially help model our multi-turn clarification process. Therefore, besides MulClari-Rule, we further propose an LLM-based method MulClari-LLM, to try to improve the multi-turn clarification effectiveness. Specifically, as for the clarification candidate generation and optimal clarification selection, we design prompts with additional information and let the LLM generate clarification candidates and select the optimal one. For the document retrieval module, we still apply BM25 for retrieval.

*3.5.1 LLM-based Clarification Candidate Generation.* The LLM-based method MulClari-LLM is illustrated in Figure 4. The process is shown on the left side, staying consistent with that in MulClari-Rule. We first use a well-designed natural language prompt to let the LLM

**Table 1: Evaluation results of document retrieval of the original query, the baseline methods, and our proposed two methods. "†" denotes that the proposed method achieves significant improvement compared with all baseline models with $p < 0.05$.**

| Model | MRR | P@1 | nDCG@1 | nDCG@5 | nDCG@20 |
|---|---|---|---|---|---|
| original $q$ | 0.1836 | 0.1027 | 0.0863 | 0.0894 | 0.0914 |
| Generation-$qD$ | 0.2114 | 0.1218 | 0.1081 | 0.1029 | 0.1012 |
| Generation-$q$ | 0.2031 | 0.1169 | 0.0963 | 0.0944 | 0.0875 |
| Labeling | 0.1848 | 0.1032 | 0.0871 | 0.0849 | 0.0735 |
| Classification | 0.1729 | 0.0903 | 0.0778 | 0.0756 | 0.0710 |
| Extraction | 0.1681 | 0.0843 | 0.0721 | 0.0697 | 0.0644 |
| MulClari-Rule | $0.2286^{\dagger}$ | $0.1332^{\dagger}$ | $0.1241^{\dagger}$ | $\mathbf{0.1245^{\dagger}}$ | $0.1143^{\dagger}$ |
| MulClari-LLM | $\mathbf{0.2374^{\dagger}}$ | $\mathbf{0.1389^{\dagger}}$ | $\mathbf{0.1263^{\dagger}}$ | $0.1239^{\dagger}$ | $\mathbf{0.1167^{\dagger}}$ |

generate multiple dimensions of the candidate items set. The prompt first describes the form of single-turn clarification, the need for multi-turn clarification, and the task purpose of retrieving a better document list. After that, we give the model several demonstrations of clarification generation to help guide the LLM.

*3.5.2 LLM-based Optimal Clarification Selection.* After the LLM-based clarification candidate generation, the LLM can generate several dimensions of related aspect items. Since our purpose is to let MulClari-LLM select one dimension that is deemed the optimal one for retrieving better documents, we further provide the model with the top 50 retrieved documents with BM25 as pseudo relevance feedback. We let the LLM select one clarification from the generated candidates to deliver to the user as shown in Figure 4.

## 4 EXPERIMENTS

### 4.1 Evaluation Data

For multi-turn clarification, we use the Qulac dataset [1] to evaluate the document ranking results. This dataset contains 198 ambiguous or faceted queries, each with corresponding labels for related and unrelated documents. For the evaluation of the quality of the first-turn clarification pane, we use a subset of MIMICS [43] to evaluate the quality of aspect items. In fact, our experiments essentially combine the advantages of two mainstream clarification datasets Qulac and MIMICS. The advantage of Qulac is that its form is closer to human dialogue, and there are corresponding annotations for relevant and irrelevant documents for each query, which is convenient for evaluation. The advantage of MIMICS is that it consists of a large number of real-world queries sampled from a search engine, making it more suitable for Web Search.

### 4.2 Evaluation Metrics

For multi-turn clarification, the effectiveness is measured by considering the performance of retrieval after updating the user query. Following existing studies [1], we apply several groups of evaluation metrics to evaluate the document ranking results, including (1) mean reciprocal rank (MRR), (2) precision of the top 1 retrieved document (P@1), and (3) normalized discounted cumulative gain for the top 1, 5, and 20 retrieved documents (nDCG@1, nDCG@5, nDCG@20). The three groups of evaluation metrics are important in different search scenarios, including traditional search engine (MRR, nDCG@5, and nDCG@20) and conversational search system with limited screen (P@1 and nDCG@1).

Furthermore, we also evaluate the quality of the first clarification pane generation. Therefore, we use four sets of single-turn evaluation metrics widely used in existing studies [11, 12, 26] to evaluate the generated aspect items. (1) Term overlap (Precision, Recall, and F1): the term overlap score evaluate the lexical similarity between the generated items and ground-truth items by comparing their same terms. (2) Exact match (Precision, Recall, and F1): the exact match score evaluates whether the generated items and the ground-truth items are totally the same. (3) Set BLEU (1, 2, 3, and 4) scores: the BLEU score calculates the n-gram overlap between two sets of texts. It is widely applied in various NLP tasks. (4) Set BERT (Precision, Recall, and F1) score: the Set BERT score calculate the similarity between two sets of texts from a semantic perspective, which make up for the shortcomings of the previous three metrics.

### 4.3 Baseline Methods

For multi-turn clarification, we implement four types of PLM-based [7, 17, 34] single-turn clarification generation approaches [26] to obtain clarification panes, including generation, labeling, classification, and extraction. The four approaches are trained with MIMICS dataset with different paradigms, and they perform well in single-turn Web search clarification. To extend them to fit the multi-turn clarification setting, after the user clicks one aspect item, the updated query will be used independently to retrieve a new document list and to generate a new clarification pane.

For evaluating the first clarification pane, we only evaluate the generated aspect items. This is because, in our proposed methods, the quality of the clarifying question is determined by the aspect items [42]. Besides, existing clarifying question generation methods have been good enough for generating clarifying questions that are not necessary for evaluation. For aspect items, we also apply the four well-performing approaches mentioned above as baselines.

### 4.4 Implementation Details

For our evaluation data, we obtain Qulac[1] and MIMICS[2] from their websites respectively. We also obtain the annotation of the document relevance of Qulac for evaluating the document ranking results as well as the top-10 search snippets of each query in MIMICS for enhancing the query. For the BART model in Section 3.4.1 and the baseline models to be compared, we optimize the BART-base

---

[1]Qulac dataset: https://github.com/aliannejadi/qulac
[2]MIMICS dataset: https://github.com/microsoft/MIMICS

**Table 2: Evaluation results for items generation. The best result for each metric is marked in bold. "†" denotes that the proposed method achieves significant improvement compared with all baseline methods with $p < 0.05$.**

| Model | Term Overlap | | | Exact Match | | | Set BLEU | | | | Set BERT | | |
|---|---|---|---|---|---|---|---|---|---|---|---|---|---|
| | Prec | Recall | F1 | Prec | Recall | F1 | 1-gram | 2-gram | 3-gram | 4-gram | Prec | Recall | F1 |
| Generation-$qD$ | 0.1423 | 0.1457 | 0.1440 | 0.0936 | 0.0912 | 0.0924 | 0.2147 | 0.1885 | 0.1724 | 0.1623 | 0.5333 | 0.5395 | 0.5364 |
| Generation-$q$ | 0.1351 | 0.1375 | 0.1363 | 0.0875 | 0.0912 | 0.0893 | 0.2084 | 0.1816 | 0.1686 | 0.1510 | 0.5351 | 0.5328 | 0.5339 |
| Labeling | 0.1615 | 0.1833 | 0.1717 | 0.1024 | 0.1275 | 0.1136 | 0.2192 | 0.1897 | 0.1767 | 0.1622 | 0.5371 | 0.5338 | 0.5354 |
| Classification | 0.0938 | 0.0956 | 0.0947 | 0.0512 | 0.0584 | 0.0546 | 0.0849 | 0.0766 | 0.0662 | 0.0608 | **0.5415** | 0.5382 | 0.5398 |
| Extraction | 0.1034 | 0.1522 | 0.1231 | 0.0463 | 0.0531 | 0.0495 | 0.2065 | 0.1771 | 0.1633 | 0.1529 | 0.5369 | 0.5413 | 0.5391 |
| MulClari-Rule | 0.1528 | 0.2527† | 0.1904† | 0.0398 | 0.0575 | 0.0470 | 0.2113 | 0.1825 | 0.1544 | 0.1323 | 0.5318 | 0.5359 | 0.5332 |
| MulClari-Rule-Best | **0.3268†** | **0.4129†** | **0.3648†** | **0.1081** | **0.1925†** | **0.1412†** | **0.3408†** | **0.2803†** | **0.2592†** | **0.2410†** | 0.5413 | **0.5399** | **0.5405** |
| MulClari-LLM | 0.0803 | 0.0885 | 0.0842 | 0.0086 | 0.0079 | 0.0082 | 0.1053 | 0.0764 | 0.0524 | 0.0389 | 0.5277 | 0.5302 | 0.5287 |
| MulClari-LLM-Best | 0.1414 | 0.1726 | 0.1554 | 0.0622 | 0.0814 | 0.0705 | 0.2173 | 0.1862 | 0.1689 | 0.1557 | 0.5359 | 0.5332 | 0.5346 |

model[3] with AdamW optimizer with the learning rate of $1.0 \times 10^{-4}$ and the batch size of 32. We hold out 10% of the MIMICS data as a validation dataset. Deep learning libraries including PyTorch and Transformers are used for training, beam searching, and validation. In the training, validation, and evaluation for item generation, we remove the item terms in the MIMICS dataset that overlap with the query terms. For example, for the query "watches" and one of its corresponding items "rolex watches", we modify the item as "rolex". We conduct this to ensure the consistency of the output. For the LLM, we use the GPT-3.5-Turbo[4].

## 4.5 Experimental Results

### 4.5.1 Multi-turn Clarification Evaluation.
We first evaluate whether the documents returned after $k$ clarification turns are more satisfactory to the users. In this section, we first set $k = 2$, which is about the "inflection point" value of the clarification turns. In other words, after more than two turns of clarification, the improvement rate of document retrieval performance slows down. We also conduct experiments with the increase of the turn $k$, which will be discussed in Section 4.7. In addition, since a clarification pane contains multiple candidate items for users to click, and clicking each candidate item will retrieve a different list of documents, we concatenate each candidate item and query provided by each clarification pane to generate a new query retrieved document list, and average the evaluation metrics generated by these document lists to obtain the final score of the current clarification pane. For the multi-turn situation ($k > 1$), we consider all possible combination of the clarification paths and select the optimal one as the final evaluation result.

Table 1 presents the results of the document retrieval after two turns of clarification. The Generation-$qD$ and Generation-$q$ mean the input is composed of the query $q$ and the snippets $D$, and only the query $q$ respectively. We can conclude from the results that, (1) **First**, most of the baseline models and our proposed methods perform better than the original query in retrieving documents after two turns of clarification. This confirms that search clarification plays an important role in the Web search, which can help the user find its satisfied documents. (2) **Second**, our proposed methods MulClari-Rule and MulClari-LLM outperform all baseline models significantly with $p < 0.05$. This result demonstrates our main conclusion: compared to existing single-turn clarification methods,

our proposed multi-turn strategy based on human-designed rules or LLMs is more suitable for clarifying users' ambiguous or faceted intent in Web conversational search. (3) **Third**, compared with MulClari-Rule, the LLM-based model MulClari-LLM achieves better results in most of the evaluation metrics. As discussed in Section 3.5, MulClari-Rule is not good at modeling multi-turn interactions, while MulClari-LLM is suitable to model the multi-turn process essentially. Therefore, it shows better performance than MulClari-Rule in a multi-turn document retrieval setting.

### 4.5.2 First-turn Clarification Pane Evaluation.
Our above experiments have shown that compared to the existing single-turn Web Search Clarification methods, our proposed multi-turn scenario and method can retrieve a more satisfactory list of documents for users. However, in addition to evaluating the quality of retrieving documents in multi-turn scenarios, we are also interested in ensuring the quality of the generated clarification candidates, especially the items. To achieve this, we evaluate the first-turn clarification quality because the first-turn clarification quality significantly determines the quality of subsequent turns and plays a very important role at the beginning. We want to answer two questions: (1) Is the dimension of multi-turn selection consistent with the ground truth in existing real-world single-turn datasets (such as MIMICS)? In other words, we want to understand whether the goal of our multi-turn method is consistent with that of single-turn methods and data. (2) Since we first generated multi-dimensional clarification candidates, we would like to observe whether these candidates include ground truth contained in single-turn datasets, even if it is not selected as optimal clarification to deliver to the user.

The experimental results are shown in Table 2. We can summarize from the result table that, (1) Compared to MulClari-LLM, the clarifications selected by MulClari-Rule are usually more close to the MIMICS dataset, showing higher performance in most of the evaluation metrics. However, MulClari-Rule's performance in multi-turn document retrieval (see Table 1) is not as good as the improvement brought by MulClari-LLM. This indicates that the single-turn clarification generation in existing studies is not as effective as our proposed model in improving document retrieval ability in multi-turn scenarios. In other words, the target in the multi-turn scenario is different from it in the single-turn scenario. (2) We record the best item dimension results of the multi-dimensional items generated by MulClari-Rule and MulClari-LLM as MulClari-Rule-Best and MulClari-LLM-Best respectively. It can be seen that

---
[3]BART-base: https://huggingface.co/facebook/bart-base
[4]GPT-3.5-turbo: https://platform.openai.com/playground?model=text-davinci-003

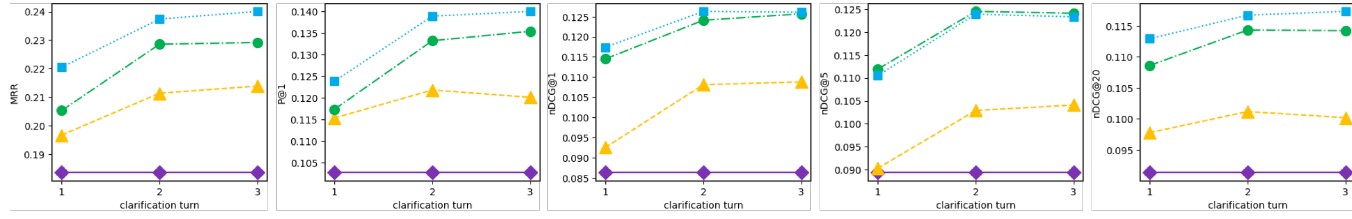

**Figure 5: Performance comparison with the baselines for different numbers of clarification turns.**

**Table 3: Our additional experimental results for MulClari-LLM. "w/o." in the table means "without".**

| Model | MRR | P@1 | nDCG@1 | nDCG@5 | nDCG@20 |
|-------|-----|-----|--------|--------|---------|
| LLM | 0.2374 | 0.1389 | 0.1263 | 0.1239 | 0.1167 |
| LLM-com | 0.2313 | 0.1347 | 0.1246 | 0.1277 | 0.1118 |
| w/o. $E$ | 0.2062 | 0.1263 | 0.0992 | 0.1016 | 0.0932 |
| w/o. $D$ | 0.2136 | 0.1127 | 0.1015 | 0.1003 | 0.0951 |

compared to the optimal clarification pane selected by the model for delivering to the users (MulClari-Rule and MulClari-LLM in Table 2), the best clarification panes (items) show great improvement in various metrics. This indicates that there are also many items corresponding to the ground truth in the MIMICS dataset that are included in the clarification candidates generated by our method. However, these items were not selected to deliver to the user in the optimal clarification selection step.

### 4.6 Additional Experiments for MulClari-LLM

The above experiments have illustrated the effectiveness of our proposed LLM-based methods and MulClari-LLM. However, some details still have not been discussed. In this section, we conduct some additional experiments to explore some details contained in MulClari-LLM. First, the clarification candidate generation and optimal clarification selection are two separate processes. In fact, in MulClari-LLM, we can combine the two processes as one whole process by modifying the prompt and letting the LLM output the best clarification pane without generating multi-dimensional candidates. The result is noted as "LLM-com" in Table 3. It is found that all metrics show a slight decrease. It proves that generating clarification candidates first is important and effective. Besides, we are also interested in the effectiveness of the demonstrations and the retrieved documents. Therefore, we remove these two modules respectively and report their results in Table 3 as "w/o. $E$" and "w/o. $D$" respectively. We see that, after removing the demonstrations, the performance shows a significant decrease, confirming that the demonstrations are important for LLMs to complete multi-turn clarification tasks. However, the top retrieved documents $D$ are not that important for LLMs, which just show a slight decrease.

### 4.7 Experiments for Clarification Turns

The performance of document retrieval is related to the specificity of the query, while the specificity of the query is related to the clarification turn, as shown in Figure 1. Therefore, the clarification turn affects the retrieval performance. Figure 5 shows the retrieval performance of our proposed methods as well as the baselines in different clarification turn $t \in \{1, 2, 3\}$. It is obvious that almost

all metrics increase with the increase of $t$. However, when the turn $t$ increases from 1 to 2, the increment is more significant than the increment when the turn $t$ increases from 2 to 3. This indicates that the previous turns (like $k = 1$ or $k = 2$) are more meaningful for clarifying user's intent. Similarly, some methods also show decrease in some evaluation metrics when $k = 3$. This proves that the clarification effect does not necessarily increase with the number of turns. Some irrelevant documents can be wrongly retrieved when the length of the user query is long. This inspire us that, in the future, it is helpful to study how to automatically determine when to **stop** clarification and only return documents in multi-turn conversational Web search.

## 5 CONCLUSION

Multi-turn clarification has been applied in various kinds of conversational search systems. However, multi-turn Web search clarification is still not comprehensively studied. In this paper, we try to extend the framework, process, and concepts of existing multi-turn clarification systems to the Web search for clarifying users' ambiguous or faceted search intents actively. We first define three important components of multi-turn Web search clarification including clarification candidate generation, optimal clarification selection, and document retrieval. Based on the framework, we design a rule-based method MulClari-Rule to generate clarification candidates and select the optimal clarification based on the frequency information of the items, and then design an LLM-based method MulClari-LLM by utilizing the in-context learning and zero-shot instruction ability of LLMs, which further improves the effectiveness of multi-turn Web search clarification. The evaluation results on the Qulac and MIMICS datasets show that, first, our proposed methods achieve better performance in improving the document retrieval ability compared with existing single-turn clarification generation methods. Second, our proposed methods can also ensure the quality of generated clarification panes. We conduct some additional experiments to further illustrate our conclusions.

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
