# OpenReview forum: "Generating Multi-turn Clarification for Web Information Seeking"
_ACM.org/TheWebConf/2024/Conference — TheWebConf24 Oral_

### Official Review · Reviewer_LePC · 2023-11-13

**Novelty:** 4
**Technical Quality:** 3

**Review:**

**Summary**: This paper extends the multi-turn clarification generation to Web search for clarifying users’ ambiguous or faceted intents. Web search queries are more complicated than conversational search scenarios and so on, so clarification should be generated. To this end, this paper first defines the whole process of multi-turn Web search clarification composed of clarification candidate generation, optimal clarification selection, and document retrieval. This paper designs a rule-based method to fit the above three components. By utilizing the in-context learning and zero-shot instruction ability of LLMs, this paper also implements clarification generation and selection by prompting LLMs with a few demonstrations and declarations, further improving the clarification effectiveness. In the experiment part, this paper validated the effectiveness of the method in improving the ability to retrieve documents and the quality of generated aspect items using Qulac and MIMICS datasets, respectively.

**Pros**:
1. This paper extends the single-turn Web search clarification to multi-turn, enriching the existing conversational search scenarios.
2. The method proposed in this paper is simple but effective, and it is easy to implement.
3. The method proposed in this paper works well and has achieved great improvement in results. This paper also proves that the method is effective on both traditional language models and LLMs.

**Cons**:
1. The difference between “conversational search” and “Web search” confuses readers. The major contribution of this work is that they change the single-turn clarification task into a multi-turn version with the help of large language models. If this work focuses on the differences between “conversational search” and “Web search”, the authors may list the different points one by one in the context.
2. The proposed method of work relies heavily on the performance of the large language model (e.g. GPT-3.5-Turbo). It is still not clear how the performance of large language models affects the multi-turn clarification task.
3. The evaluation methods should involve efficiency and human evaluations. Only providing performance metrics does not verify the effectiveness of the proposed method because the user action introduces more information into the model which definitely increases the overall relevant performance.
4. The user click simulation is not described in the paper. How to determine which selections the user should click in the current turn and is it possible that the user does not like any of the provided selections?
5. The writing problem should draw much attention.
 - Table 2 the last second column wrong bold font mark.
 - The figures in the paper are not clear.
 - No legend in Figure 5.

**Questions:**

1. The authors assert that they are pioneers in expanding the single-turn Web search clarification to multi-turn scenarios. However, there is some ambiguity, and it remains uncertain whether this claim might be overstated. The techniques employed in CRS can also be considered as multi-turn baselines. Could the authors provide further clarification to explicitly highlight the novelty of being the first to propose multi-turn Web search clarification?

2. How does the proposed method fare when applied to various types of LLMs, such as vicuna or lamma2?

3. What are the efficiency metrics used to evaluate the proposed method, and are there any human-centric metrics suggested to validate its performance?

4. What is the methodology for simulating user clicks?

**Reviewer Confidence:**

3: The reviewer is confident but not certain that the evaluation is correct

**Scope:**

4: The work is relevant to the Web and to the track, and is of broad interest to the community

---

### Official Review · Reviewer_H29q · 2023-11-22

**Novelty:** 6
**Technical Quality:** 5

**Review:**

The paper is well-written and easy to follow. It proposes a new multi-turn clarification approach to be applied to search engines,
similar to what has been done with Conversational Search recommendation systems. The authors clearly explain the opportunities this approach provides in a search-engine approach. The idea of adding multiple turns of clarification is indeed helpful, as shown in the paper. They also exploit generative a and few-shot learning capabilities of gpt3.5-turbo to increase the retrieval phase's quality further.

There are only some points that should be better explained; for instance, after selecting the first clarification pane, a new retrieval phase is performed, which means that computationally, it would be more costly than a single-turn clarification approach. Also, the results obtained in Table 1 show the results obtained after two clarification turns; it would be interesting to show how the MulClari in the two configurations performs after only one turn. The state-of-the-art comparisons were devised to be used with one turn only. Another note is the captions of the tables, which should be more self-explanatory.

Overall, the paper inserts well in a research area such as recommendation systems, providing a new approach and showing
increased results compared to the state-of-the-art. The experiments are well-described and motivated. The quality of the paper is good. As far as my knowledge in the area goes, the paper should be accepted.

**Questions:**

The paper is clear and well-written. The only issue I had while reading regards the multiple retrieving phases when elaborating clarification turns that may result in a considerable increase of computational time/cost. Have you considered addressing this issue?

Also, when comparing with SOTA approaches, you apply them in a two-step scenario, it would be interesting to see how you perform in a single-step setting. In this way, the significance of the multi-turn approach would be clearer.

A last suggestion for the future regards the LLM. In the paper, it seems that you used GPT3.5-turbo instructed. Have you considered trying finetuning a much smaller model for the task? In this way, you may avoid using prompting, which is changeable, and you could save a lot of computational costs.

**Ethics Review Description:**

does not apply

**Reviewer Confidence:**

3: The reviewer is confident but not certain that the evaluation is correct

**Scope:**

4: The work is relevant to the Web and to the track, and is of broad interest to the community

---

### Official Review · Reviewer_VwSA · 2023-11-24

**Novelty:** 5
**Technical Quality:** 6

**Review:**

**Summary:**
The paper presents a novel framework that extends traditional single-turn web search to a multi-turn system. This framework aims to assist users in efficiently searching for documents by minimizing the number of clarification turns required. To demonstrate the framework's efficacy, the authors developed a rule-based method, MulClari-Rule, and a Large Language Model (LLM)-based approach. These methods have shown to significantly improve document retrieval abilities compared to existing single-turn clarification generation methods, while also maintaining the quality of the generated clarification panes.

**Strengths:**
1. **Innovative Multi-Turn Web Search**: The transition from single-turn to multi-turn web search is a notable advancement, enhancing the search experience.
2. **Efficiency in Document Search**: The framework's design focuses on reducing the number of clarification turns, thereby streamlining the search process for users.
3. **Effective Rule-Based Method**: The introduction of MulClari-Rule demonstrates the practical effectiveness of the framework.
4. **Versatile LLM-Based Approach**: The implementation of a Large Language Model-based approach shows the adaptability and modern application of the framework.
5. **Enhanced Document Retrieval Performance**: The proposed methods outperform existing single-turn clarification generation methods in document retrieval.
6. **High-Quality Clarification Panes**: The framework ensures the production of high-quality clarification panes, crucial for user interaction and satisfaction.

**Limitations and weakness:**
1. **Unclear Differences between Conversational Search and Web Search**: The distinction between 'conversational search' and 'Web search' leads to confusion. This work's contribution is transforming the single-turn clarification task into a multi-turn version, facilitated by using large language models. To better focus on the differences between 'conversational search' and 'Web search,' it would be advantageous if the authors systematically listed and discussed these differences within their context.

2. **Unclear Clarification Termination Guidelines**: The paper does not discuss how or when the clarification process should be terminated. Providing a strategy or algorithm would significantly improve the framework's usability and effectiveness.

**Questions:**

none

**Reviewer Confidence:**

3: The reviewer is confident but not certain that the evaluation is correct

**Scope:**

4: The work is relevant to the Web and to the track, and is of broad interest to the community

---

### Official Review · Reviewer_ZdYb · 2023-11-24

**Novelty:** 6
**Technical Quality:** 6

**Review:**

The paper presents a novel strategy to generate clarification panes that can be used by the user, which can click on the items in the clarification pane, to improve and clarify their query.

The paper is overall clear and well motivated.

As a weak aspect that i noticed, the mathematical formulation section 3.2 should be clarified better. Most of the mathematical objects are not defined and need to be subsumed. For example, what is the meaning of Di? is it the i-th document retrieved or the i-th documents set, after i turns of clarifying the question? what are Qi and Si? what is Phi?

Additionally, "item" does not seem the most appropriate term: from my understanding, it is a word/facet/concept. Using the term item is a bit confusing, as they are not "items".

**Questions:**

As a weak aspect that i noticed, the mathematical formulation section 3.2 should be clarified better. Most of the mathematical objects are not defined and need to be subsumed. For example, what is the meaning of Di? is it the i-th document retrieved or the i-th documents set, after i turns of clarifying the question? what are Qi and Si? what is Phi?

**Ethics Review Description:**

No issue

**Reviewer Confidence:**

3: The reviewer is confident but not certain that the evaluation is correct

**Scope:**

4: The work is relevant to the Web and to the track, and is of broad interest to the community

---

### Official Review · Reviewer_3JBY · 2023-11-26

**Novelty:** 4
**Technical Quality:** 4

**Review:**

The paper addresses the creation of clarifying questions for web search. Evaluation is in terms of similarity of generated clarifications viz. some ground truth dataset, and in terms of document ranking effectiveness.

Overall, while the approach has some merit, the paper needs to better signpost what is being done by clearly explaining the notation and the experimental methodology. I am familiar with the area, but I struggled to follow the paper. Some explicit points:

 - What are Q_i and S_i? They are never defined/described explicitly. What is an aspect Item?

 - Its unconventional that the propsoed framework is described as using BART, when there are many generative models. Why not leave it abstract and then experiment with the choice of generative model later?

 - Tense is varied, including in the related work (mixing present and past tence); the method section also dances between present and past and also using ("we can")

 - what are the robust clarifying question generation algorithms you mentino at line 496.

 - what is H(D|q) etc. The meaning of H function is not clear.

 - Section 4.4 tells us that QUlac and MIMCS are obtained from their websites, but not what they are actually used for. I'm really missing the big picture. How are generations actually evaluted, what is the ground truth (ie which dataset it comes from) to use the BLEU etc metrics defined in Section 4.2

 - Section 5 (Conclusions) omits any statement of the findings/conlusions. Also it says "we try to extend the framework" -- were you not successful?

Minor comments:

 - line 109: How can a clarification be restarted
 - what is a pane? its only really defined in Figure 1, but not in the text
 - line 224: what is "it"?
 - generally the tense in the related work section is a mix of present and past.
 - please use \langle and \rangle for tuples, rather than () and <> - you use both (line 347, line 442).
 - line 574: "we still apply" -- why "still"?
 - lines 856, 887: we cannot prove anything experimentally.
 - Section 4.6: I cannot connect with w/o E., w/o D - what does these acronyms refer to?

**Questions:**

See comments above.

**Reviewer Confidence:**

3: The reviewer is confident but not certain that the evaluation is correct

**Scope:**

3: The work is somewhat relevant to the Web and to the track, and is of narrow interest to a sub-community

---

### Decision · Program_Chairs · 2024-01-22

**Decision:**

Accept (Oral)

**Comment:**

This paper addresses the creation of clarifying questions for web search.

 The paper was reviewed by five reviewers. The paper has clearly some merits. Most reviewers agree on the technical quality and novely of the papers, but they also raise some comments still requiring a proper explanation. In particular, the actual content must be improved to allow readers to position and understand the paper w.r.t. the existing works, using proper, commonly adopted terminology in the QA research community.